# COVID-19 and vaccine hesitancy: A longitudinal study

**Ariel Fridman** [ID]*, **Rachel Gershon, Ayelet Gneezy**

Rady School of Management, University of California San Diego, La Jolla, California, United States of America

* afridman@ucsd.edu

**Data Availability Statement:** All data and code are publicly available on the Open Science Framework at https://osf.io/kgvdy/.

**Funding:** UC San Diego Global Health Initiative (GHI): awarded to all authors; Project number: 1001288. The funders had no role in study design,

## Abstract

How do attitudes toward vaccination change over the course of a public health crisis? We report results from a longitudinal survey of United States residents during six months (March 16 –August 16, 2020) of the COVID-19 pandemic. Contrary to past research suggesting that the increased salience of a disease threat should improve attitudes toward vaccines, we observed a decrease in intentions of getting a COVID-19 vaccine when one becomes available. We further found a decline in general vaccine attitudes and intentions of getting the influenza vaccine. Analyses of heterogeneity indicated that this decline is driven by participants who identify as Republicans, who showed a negative trend in vaccine attitudes and intentions, whereas Democrats remained largely stable. Consistent with research on risk perception and behavior, those with less favorable attitudes toward a COVID-19 vaccination also perceived the virus to be less threatening. We provide suggestive evidence that differential exposure to media channels and social networks could explain the observed asymmetric polarization between self-identified Democrats and Republicans.

## Introduction

Vaccinations are among the most important public health tools for reducing the spread and harm caused by dangerous diseases [1]. The World Health Organization estimates that vaccines prevented at least 10 million deaths between 2010–2015 worldwide [2]. Despite considerable evidence showing vaccines are safe [3, 4], there is increasing skepticism toward vaccination [5, 6]. Vaccine hesitancy has led to a decline in vaccine uptake and to an increase in the prevalence of vaccine-preventable diseases (VPDs) [7, 8]. Ironically, the objection to vaccines is commonly a consequence of their effectiveness—because individuals have lower exposure to VPDs, they are less concerned about contracting them [9], which consequently leads to greater vaccine hesitancy [10]. The COVID-19 pandemic has created a new reality where individuals are faced with a previously unknown disease and its effects, providing a unique opportunity to investigate vaccine attitudes during a period of heightened disease salience. The present research reports findings from a longitudinal study conducted during the COVID-19 health crisis, in which we measured changes in attitudes toward a prospective vaccine, as well as shifts in vaccine attitudes in general.

data collection and analysis, decision to publish, or preparation of the manuscript. https://medschool. ucsd.edu/som/medicine/divisions/idgph/research/ Global-Health/grant-recipients/2019-2020/Pages/ Faculty-Postdoc-Travel-and-Research.aspx.

**Competing interests:** The authors have declared that no competing interests exist.

## Factors influencing vaccine attitudes and behaviors

Past research has identified a variety of situational and individual-level factors that influence vaccine attitudes and behavior, the most prominent of which are risk perceptions and demographic characteristics.

Assessments of risk are influenced by both cognitive evaluations (i.e., objective features of the situation such as probabilities of outcomes) and affective reactions [11], as well as by contextual factors (e.g., the information that is most available or salient at the time [12]). For example, research shows that media coverage plays a significant role in determining the extent to which we take threats seriously [13]. When individuals perceive heightened risk of a threat, they become more favorable toward interventions that mitigate that threat, including vaccination (for a meta-analysis on the effect of perceived risk on intentions and behaviors, see [14]). In the case of COVID-19, this would suggest more positive attitudes toward a vaccine and greater likelihood to get vaccinated. Indeed, research suggests that individuals should exhibit a greater interest in vaccinations during a pandemic because disease threat is more salient [15].

Past efforts to improve vaccine attitudes have had limited success or even backfired; for example, messages refuting claims about the link between vaccines and autism, as well as messages featuring images of children who were sick with VPDs, had negative effects on vaccine attitudes among those who were already hesitant to vaccinate [16]. In contrast, messaging that increases disease threat salience has shown promise in reducing vaccine hesitancy [5], and there is evidence suggesting that increased threat salience for a particular disease may also increase intentions to vaccinate for other diseases [17]. Building on these findings, we expected to find an increase in pro-vaccine attitudes and in individuals' interest in a COVID-19 vaccine when the perceived threat of the COVID-19 virus increased.

Vaccine attitudes are also influenced by a variety of demographic and ideological factors (for a review, see [18]). For example, perceptions of vaccine risk differ among individuals of different ethnic backgrounds [19], and there is extant work demonstrating a positive correlation between socioeconomic status (SES) and vaccine hesitancy [20, 21]. Socio-demographic factors are also linked to vaccine-related behaviors: among college students, those whose parents have attained a higher level of education are more likely to get immunized [22], and researchers have identified age as a predictor for receiving the influenza vaccine [23].

Political ideology is another well-documented determinant of vaccine-related attitudes and behaviors. Despite a common belief that liberals tend toward anti-vaccination attitudes in the United States, there is strong evidence that this trend is more present among conservatives [24, 25]. According to a recent Gallup Poll, Republicans are twice as likely to believe the widely debunked myth that vaccines cause autism [26]. Recent work has shown that exposure to anti-vaccination tweets by President Trump—the first known U.S. president to publicly express anti-vaccination attitudes—has led to increased concern about vaccines among his supporters [27]. Based on these findings, and in conjunction with the sentiments expressed by the White House that diminished the significance of the pandemic [28], we expected to find diverging trends between Democrats and Republicans.

## The current research

We collected vaccine-related attitudes of individuals living in the U.S. over a six-month period. Beginning in March 2020, we elicited attitudes from a cohort of the same individuals every month. We began data collection before any COVID-19 lockdown measures were in place (i.e., prior to the nation's first shelter-in-place order [29]). Hence, our data spans the early phase of the pandemic, when there were fewer than 2,000 total confirmed cases in the

U.S., through the following six months, at which point cumulative cases reached over 5.3 million [30].

Despite our prediction—that a public health crisis would increase disease threat, consequently increasing pro-vaccine attitudes and interest in vaccination—our data show an overall decrease in favorable attitudes toward vaccines. A closer look at the data revealed that political orientation explains more variance than any other socio-demographic variable. Specifically, participants who identify as Republican showed a decrease in their intention to get the COVID-19 vaccine and the influenza vaccine as well as a general decrease in pro-vaccine attitudes, whereas Democrats' responses to these measures did not show a significant change during this period.

Our work is the first, to our knowledge, to longitudinally measure individuals' attitudes toward vaccines. In doing so, our findings advance the understanding of how vaccine attitudes might change during an unprecedented public health crisis, revealing a strong association between political party affiliation and vaccine attitudes.

## Methods

### Participants

We recruited a panel of U.S. residents on Amazon's Mechanical Turk platform to respond to multiple survey waves. To incentivize completion of all waves, we informed participants their payment would increase for subsequent surveys. Participants were paid 30 cents for wave 1, 40 cents for wave 2, and 60 cents for waves 3 and 4, $1.00 for wave 5, and $1.20 for wave 6. In addition, participants were informed that those who completed the first three waves would enter a $100 raffle. The median survey completion time was 5.5 minutes. The sample size for the first wave was 1,018, and the number of participants ranged from 608–762 on subsequent waves (see S1 Table for attrition details). This project was certified as exempt from IRB review by the University of California, San Diego Human Research Protections Program (Project #191273XX).

Our panel represents the broad and diverse population of the United States. The first wave sample included participants from all 50 states (except Wyoming) and Washington D.C., with an age range of 18 to 82 years old (mean = 38.48, median = 35). Approximately half (53%) identified as male, 46% as female, and .6% as other. The racial makeup in our sample was: 80% White, 9% Asian, 6% Black or African American, 4% multiple racial or ethnic identities, and 1% other. Relative to the U.S. Census (2019) [31] estimates, our sample over-represents White and Asian individuals, and under-represents Black or African American individuals and other racial groups.

We elicited political affiliation using a 6-point Likert scale, ranging from Strongly Republican to Strongly Democratic. In wave 1, 62% identified as Democrats and 38% identified as Republican, which is consistent with results from the most recent General Social Survey (GSS) [32]. There was no significant change in mean political identity from wave 1 to waves 2–6 (see S2 Table). We classified participants as Democrats or Republicans using wave 1 political party affiliation. See S2 Appendix for additional details about the correlation of political party affiliation with age, gender, and SES.

### Questions and measures

Our primary measure of interest was participants' stated intention to get the COVID-19 vaccine when it becomes available. We were also interested in their perceptions of COVID-19 threat, general vaccination attitudes, and intention to get the flu shot. For all measures, except flu shot intentions, we combined multiple items to create a composite measure (see S2 Table

for specific questions and construct compositions). Questions designed to measure general vaccination attitudes were adapted from prior work [33].

Additional measures of interest were participants' trust in broad institutions (media, local government, and federal government). These trust measures followed different trends from each other, and therefore were not combined. At the end of the survey, participants responded to demographic questions. We retained all questions used in wave 1 throughout all six waves (our survey included additional items not reported in this paper; see S2 and S3 Tables for a complete list of measured items).

### Data and analysis plan

Only participants with non-missing and non-duplicated responses were included in the analyses (see S1 Appendix for additional details). For all outcomes of interest, we tested for linear trends over time using a fixed effects regression specification [34]. All regression results include individual-level fixed effects, and standard errors are clustered at the individual level, to adjust for within-person correlation. We used this approach to control for the impact of omitted or unobserved time-invariant variables. P-values are not adjusted for multiple testing (see [35]). All analyses were conducted using R (version 4.0.2), and regressions were run using the package "fixest" (version 0.6.0). All materials, data, and additional analyses including robustness checks can be found here: https://osf.io/kgvdy/.

## Results

We report results for three different vaccination-related measures: attitudes toward a COVID-19 vaccine, general vaccination attitudes, and flu shot intentions. All measures showed a decreasing trend (Ps < .001, except flu shot intentions where $p$ = .05) for the 6-month duration of the study, indicating a reduction in pro-vaccination attitudes and intention to get vaccinated (COVID-19 and influenza vaccines). See S4 Table for full results of all regressions.

### Heterogeneity in trend by political party

To better understand whether the decline in vaccine attitudes over time was driven by a particular factor, we used a data-driven approach, regressing all demographic characteristics on vaccine attitudes, in separate regressions. These demographics included education, income, SES, race, gender, an item measuring whether participants considered themselves to be a minority, whether the participant has children, and political party. Education, income, and SES were median split; race and gender were dummy coded; and political party affiliation was dichotomized into Democrat or Republican. Among all demographic characteristics, separating time trends by political affiliation (by adding an interaction term) attained the greatest adjusted within-$R^2$ in explaining vaccination attitude measures. In other words, political party affiliation explains the greatest within-individual variation in vaccine attitudes over time.

An analysis of responses by political affiliation revealed that the observed decreasing trend in all three vaccine measures was mostly driven by participants who identified as Republican (all Ps < .05), whereas Democrats' responses showed either no significant trend (for COVID-19 vaccination and flu shot intentions: Ps >.67) or a significantly less negative time trend (general vaccination: $p$ < .001). For these regressions, and moving forward, all results included interactions between wave and political party as well as interactions for wave and age, and wave and SES, to control for potentially different time trends associated with these variables. In each regression we also tested whether the strength of political affiliation moderates the observed results, and we reported the result when it did. We also conducted ANOVAs to

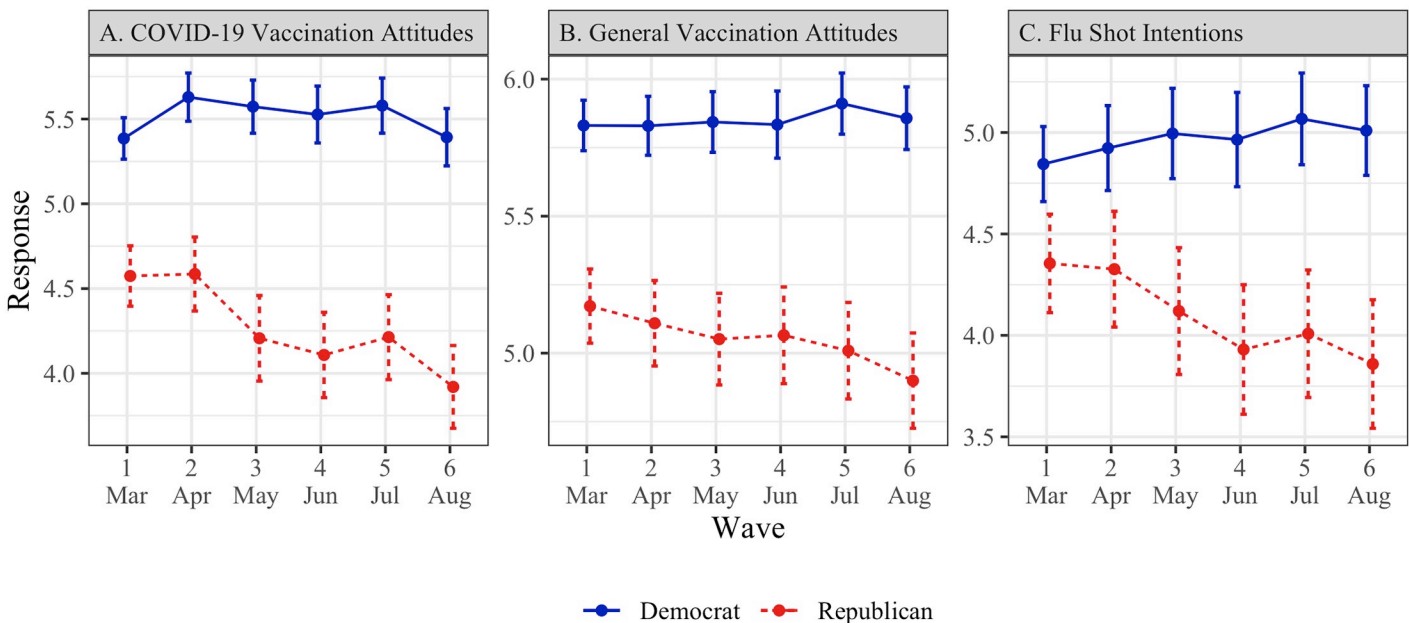

**Fig 1. Vaccination attitudes and intentions by political affiliation (March–August 2020).** Points represent means, and error bars represent 95% confidence intervals. All scale responses range from 1 to 7.

compare mean responses for the outcomes of interest between Democrats and Republicans, separately for each wave (see S5 Table).

**COVID-19 vaccination attitudes (Fig 1, Panel A).** A two-item construct ($r$ = .78) was created, with greater values corresponding to more favorable responses.

In wave 1, Democrats ($M$ = 5.39, SD = 1.55) had more favorable attitudes toward a COVID-19 vaccine than Republicans ($M$ = 4.57, SD = 1.76; $t$ = -7.38, $p$ < .001, $d$ = -.48, 95% CI = [-.61, -.35]). Among Democrats, there was no significant time trend ($\beta$ = .02, SE = .04, $p$ > .67) whereas Republicans' responses followed a decreasing time trend ($\beta$ = -.09, SE = .05, $p$ = .046). These trends were significantly different from each other ($\beta$ = -.11, SE = .02, $p$ < .001).

**General vaccination attitudes (Fig 1, Panel B).** A ten-item construct ($\alpha$ = .95) was created, with greater values corresponding to a more positive attitude toward vaccination in general.

In wave 1, Democrats ($M$ = 5.83, SD = 1.15) expressed more favorable general vaccination attitudes than Republicans ($M$ = 5.17, SD = 1.31; $t$ = -7.91, $p$ < .001, $d$ = -.52, 95% CI = [-.66, -.39]). Although both Democrats and Republicans had a decreasing time trend (Democrats: $\beta$ = -.04, SE = .02, $p$ = .029; Republicans: $\beta$ = -.09, SE = .02, $p$ < .001), the trend for Republicans was significantly more negative ($\beta$ = -.04, SE = .01, $p$ < .001).

**Flu shot intentions (Fig 1, Panel C).** We asked participants whether they plan to get the flu shot next year, with greater values indicating greater intentions.

In wave 1, Democrats ($M$ = 4.84, SD = 2.34) indicated greater intentions to vaccinate against the flu than Republicans ($M$ = 4.35, SD = 2.39; $t$ = -3.15, $p$ = .002, $d$ = -.21, 95% CI = [-.34, -.08]). Among Democrats, there was no significant time trend ($\beta$ = .01, SE = .04, $p$ = .86), suggesting their vaccination intentions remained largely stable. Republicans' responses, however, revealed a decreasing time trend ($\beta$ = -.12, SE = .04, $p$ = .005), and the two trends were significantly different from each other ($\beta$ = -.12, SE = .02, $p$ < .001).

Our analyses revealed an interaction with political affiliation strength among Republicans, whereby participants who identified as more strongly Republican had a more negative time

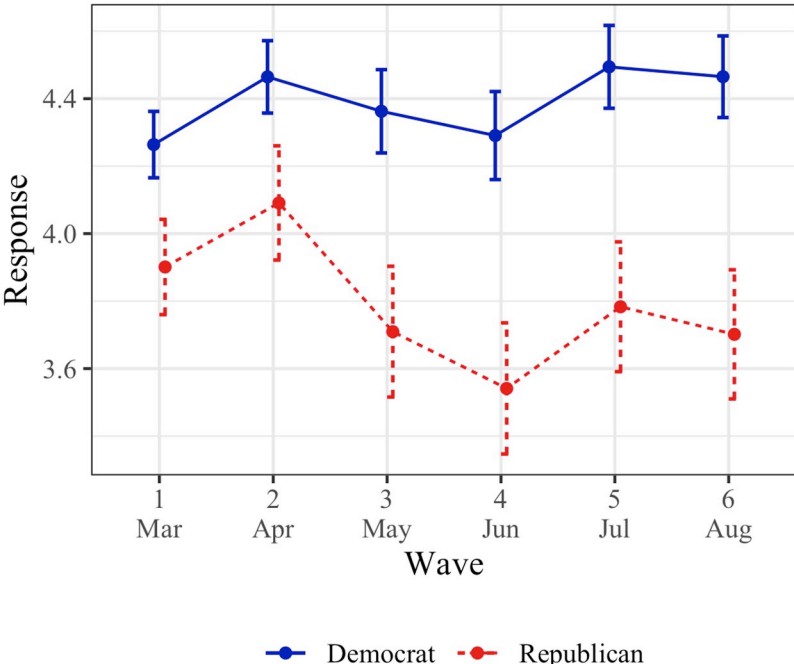

**Fig 2. Perceived threat of COVID-19 by political affiliation (March–August 2020).** Points represent means, and error bars represent 95% confidence intervals. All scale responses range from 1 to 7.

trend ($\beta$ = -.05, SE = .02, $p$ = .027). This interaction was not significant for Democrats ($\beta$ = -.02, SE = .01, $p$ = .19).

**Perceived threat of COVID-19 (Fig 2).** A three-item construct ($\alpha$ = .82) was created, with greater perceived threat about COVID-19.

In wave 1, Democrats ($M$ = 4.26, SD = 1.25) expressed greater perceived threat of COVID-19 than Republicans ($M$ = 3.90, SD = 1.39; $t$ = -4.14, $p$ < .001, $d$ = -.40, 95% CI = [-.27, -.14]). Democrats' responses showed an increasing time trend ($\beta$ = .08, SE = .04, $p$ = .033), indicating they became increasingly concerned about the threat posed by the virus over time. Among Republicans, there was no significant time trend ($\beta$ = -.01, SE = .04, $p$ = .83). These trends were significantly different from each other ($\beta$ = -.09, SE = .02, $p$ < .001). While our data does not render causal claims, it is possible that the divergence in COVID-19 threat perceptions over time among Republicans and Democrats contributes to the divergence in vaccine attitudes between these groups over time. We revisit this proposition in the General Discussion.

Our analyses revealed an interaction with political affiliation strength among Democrats—participants who identified as more strongly Democrat had a more positive time trend ($\beta$ = .03, SE = .01, $p$ = .019), suggesting an increasing threat perception over time. This interaction was not significant for Republicans ($\beta$ = .01, SE = .02, $p$ = .61).

**Trust in broad institutions.** The measures of trust in media, local government, and federal government were not highly correlated ($\alpha$ = .66), and were therefore analyzed separately.

*Trust in media (Fig 3, Panel A).* In wave 1, Democrats ($M$ = 3.61, SD = 1.66) reported greater trust in the media than Republicans ($M$ = 2.73, SD = 1.65; $t$ = -8.12, $p$ < .001, $d$ = -.53, 95% CI = [-.66, -.39]). There was no significant time trend for either Democrats ($\beta$ = .02, SE = .04, $p$ = .57) or Republicans ($\beta$ = -.05, SE = .04, $p$ = .20). However, the trend for Republicans was significantly more negative ($\beta$ = -.07, SE = .02, $p$ < .001). The different trends we observe for Democrats and Republicans with respect to trust in the media may explain the

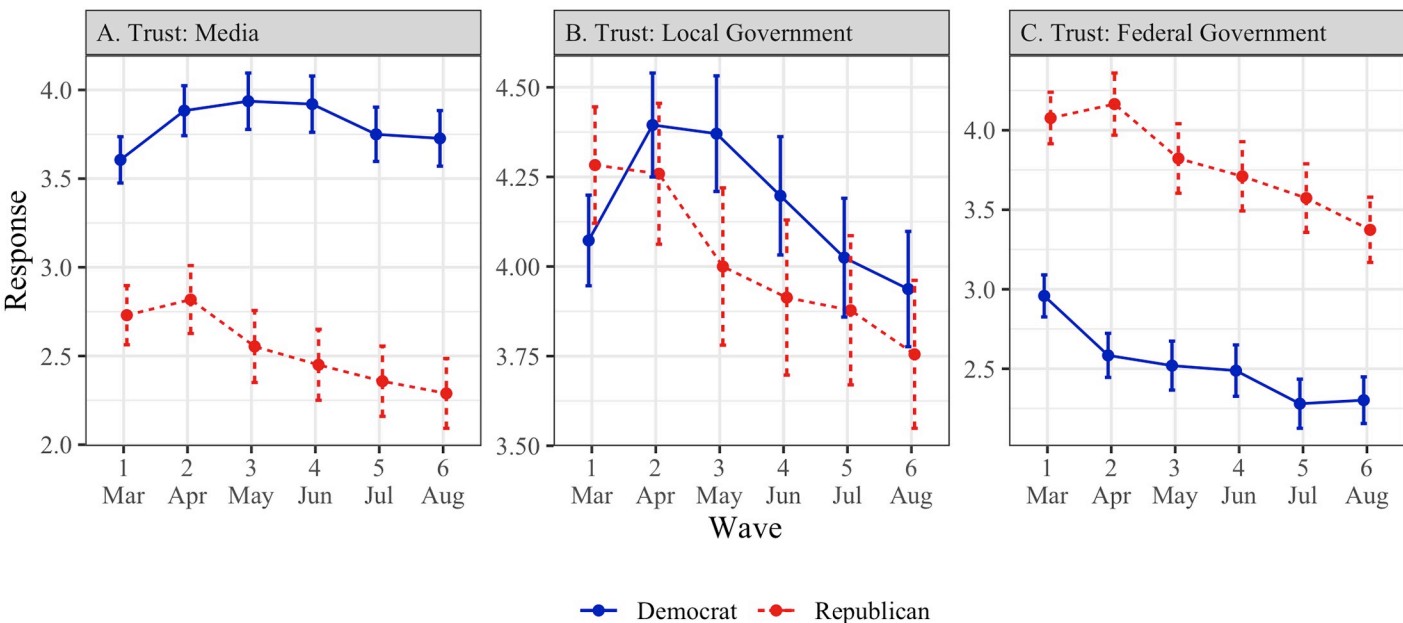

**Fig 3. Trust in broad institutions by political affiliation (March–August 2020).** Points represent means, and error bars represent 95% confidence intervals. All scale responses range from 1 to 7.

divergence in perceived threat and vaccine attitudes between these groups over time (see General discussion).

*Trust in local government (*Fig 3*, Panel B).* In wave 1, Democrats (*M* = 4.07, SD = 1.60) indicated lower trust in local government than Republicans (*M* = 4.28, SD = 1.60; *t* = 2.01, *p* = .045, *d* = .13, 95% CI = [.003,.26]). Among Democrats, there was no significant time trend (*β* = -.06, SE = .04, *p* = .18), though among Republicans, there was a decreasing time trend (*β* = -.11, SE = .05, *p* = .015). These trends were significantly different from each other (*β* = -.06, SE = .02, *p* = .004).

*Trust in federal government (*Fig 3*, Panel C).* In wave 1, Democrats (*M* = 2.96, SD = 1.67) expressed lower trust in the federal government than Republicans (*M* = 4.08, SD = 1.60; *t* = 10.52, *p* < .001, *d* = .68, 95% CI = [.55,.82]). Both Democrats and Republicans had decreasing time trends (Democrats: *β* = -.08, SE = .04, *p* = .036; Republicans: *β* = -.10, SE = .04, *p* = .025). These trends were not significantly different from each other (*β* = -.02, SE = .02, *p* = .37).

## Attrition

To rule out differential attrition, we tested whether the composition of our sample (i.e., age, gender, and political party) changed over time (see S1 Table). Specifically, we tested whether participants who responded to waves 2–6 were significantly different at baseline (wave 1) from the full sample at baseline. The only significant change detected (Ps < .05) was with respect to participants' age, though the differences were small—the average age was 38.5 at baseline, and remained between 39.9 and 40.8 at baseline among participants who responded to subsequent waves. We found no other systematic pattern of attrition among our participants.

## General discussion

Over the course of six months of the COVID-19 pandemic, beginning with a relatively early phase prior to any U.S. directives to stay home (March 2020) and continuing through a

cumulation of over 5 million cases (August 2020), we found a decrease in pro-vaccine attitudes and COVID-19 vaccination intentions, as well as reduced intentions to get the influenza vaccine. These findings are contrary to our prediction that increased salience of COVID-19 would improve attitudes toward vaccines.

Our analyses identify political ideology as the best predictor of the decreasing time trend across our three vaccine-related attitudes and intentions measures. In particular, we found that while Democrats' responses remained fairly stable over time, Republicans shifted away from their lower initial responses and from Democrats' responses, leading to increased polarization throughout the six-month period.

Contrary to the polarization observed in our data, social and behavioral scientists have long argued that groups facing threats often come together, demonstrating stronger social cohesion [36], and more cooperative behaviors [37, 38]. Researchers have also found that individuals' sense of shared identity plays a role in promoting cooperative behavior in response to threat [39–41]. Considering our results in the context of these findings might suggest that our respondents' sense of shared identity was dominated by their political ideology, as opposed to a broader (e.g. American) identity.

## What might be going on?

Although the nature of our data does not render causal claims, it highlights potential explanations. First, we note that participants' ratings of perceived COVID-19 threat followed a similar diverging pattern by party affiliation to our three vaccine-related measures during the study period. Democrats perceived COVID-19 threat to be greater at the start of the study than Republicans did, and this gap widened significantly as the study progressed. This trend is consistent with previous research showing that vaccine hesitancy is related to perceived risk of a threat; when a VPD threat level is low, individuals are less motivated to take preventative action (i.e., immunize; for a review, see [42]).

Our data offers one potential explanation for the polarization of threat perception: Republican and Democratic participants in our study reported consuming different sources of information. The most commonly checked news source for Republicans was Fox News (Republicans: 50%, Democrats: 8%; $\chi^2$ = 164.55, $p$ < .001) and for Democrats was CNN (Democrats: 47%, Republicans: 23%, $\chi^2$ = 43.08, $p$ < .001, see S6 Table). Corroborating this proposition, a Pew Research Center poll conducted in March 2020 found that 56% of respondents whose main news source is Fox News believed that "the news media have greatly exaggerated the risks about the Coronavirus outbreak," whereas this was only true for 25% of those whose main news source is CNN [43]. Of note, Facebook and Instagram, were also in the top four most consumed news sources for participants affiliated with either party. Extant work describes these platforms as echo chambers [44, 45], which may exacerbate partisan exposure to news and information.

Another trend highlighted by our data shows that similar to vaccine attitudes, Republicans' trust in the media decreased significantly more during our study than Democrats', suggesting these patterns might be related. According to Dr. Heidi Larson, an expert on vaccine hesitancy and founder of the Vaccine Confidence Project, misinformation regarding vaccinations is more likely to take root when individuals do not trust the information source [46]. Future research might further examine the role of trust in the media on vaccine attitudes.

While trust in media or media exposure may be driving COVID-19 threat perceptions and vaccine attitudes, there are many other possible explanatory factors that are not captured by our data or analyses. For example, it is possible that threat perceptions were influenced by how a respondents' county or state was affected by COVID-19; up until June 2020, COVID-19

cases were more common in Democrat-leaning states [47], which might have amplified its salience early on and influenced attitudes and behavior. Further, although we included individual-level fixed-effects which control for all time invariant participant characteristics, and controlled for different trends by age and SES, we cannot rule out the possibility that other factors (e.g., educational attainment or population density) may have influenced the observed trends. Finally, as our data collection began after the onset of COVID-19, it is possible that the trend we observe for Republicans represents a return to a pre-pandemic baseline of vaccine-related attitudes.

## Contributions

This work advances our understanding of how health-related attitudes evolve over time. Our focus on vaccine-related attitudes and intentions is important because experts agree that having enough people vaccinate against COVID-19 is key to stemming the pandemic [48]. More broadly, negative attitudes toward vaccination in general, and reduced vaccine uptake, is increasingly a public health concern [49]. This research provides insight into the trends of such vaccine hesitancy, underlining the importance of risk salience and its roots in ideology and media exposure.

This work also contributes to our understanding of political parties and polarization. Numerous anecdotes and reports have demonstrated a partisan divide in Americans' response to the COVID-19 pandemic. For example, research found greater negative affective responses to wearing a face covering among politically right (vs. left) leaning individuals [50]. Here, we show that although these observations are valid, the reality is more nuanced. For example, our analyses reveal that polarization on vaccine measures—both attitudes and intentions—is driven primarily by self-identified Republicans' gradual movement away from their initial responses whereas Democrats' responses remained largely stable. This insight has important practical implications: It informs us about the dynamics of individuals' attitudes, bringing us closer to understanding the underlying factors that influence attitudes and behaviors. Equipped with this knowledge, one could design more effective communications and interventions.

## Note on methodology and data availability

The present study contributes to a small but growing literature in the social sciences using longitudinal data [51]. Using a longitudinal methodology allowed us to track individual-level changes over time. Merely observing a single point in time would allow us to observe across-group differences, but would lack the bigger picture of how polarization between these groups evolved. Another key advantage of panel data is that it allows us to include individual-level fixed effects, which control for the impact of omitted or unobserved time-invariant variables. Finally, panel data allows for more accurate inference of model parameters [52].

While the focus of this paper is vaccine attitudes, our broad dataset offers a unique opportunity to understand attitudes and behavior over time. Due to the richness of our data, its unique nature, and its timeliness, we believe it is important to make it available to other researchers interested in exploring it and publishing additional findings. The complete dataset is available at https://osf.io/kgvdy/ (see S2 and S3 Tables for all items collected).

## Supporting information

**S1 Appendix. Additional information about sample exclusions.**
(DOCX)

**S2 Appendix. Additional information about political party affiliation.**
(DOCX)

**S1 Table. Attrition table.**
(DOCX)

**S2 Table. Summary table of measures and constructs included in the text.**
(DOCX)

**S3 Table. Summary table of measures excluded from the text.**
(DOCX)

**S4 Table. Regression results.**
(DOCX)

**S5 Table. Outcome measures by political party affiliation.**
(DOCX)

**S6 Table. Summary of news sources.**
(DOCX)

## Author Contributions

**Conceptualization:** Ariel Fridman, Rachel Gershon, Ayelet Gneezy.

**Data curation:** Ariel Fridman, Rachel Gershon.

**Formal analysis:** Ariel Fridman.

**Funding acquisition:** Rachel Gershon, Ayelet Gneezy.

**Investigation:** Ariel Fridman, Rachel Gershon, Ayelet Gneezy.

**Methodology:** Ariel Fridman, Rachel Gershon, Ayelet Gneezy.

**Project administration:** Ariel Fridman, Rachel Gershon, Ayelet Gneezy.

**Resources:** Ariel Fridman.

**Supervision:** Ariel Fridman, Rachel Gershon, Ayelet Gneezy.

**Validation:** Ariel Fridman, Rachel Gershon.

**Visualization:** Ariel Fridman, Rachel Gershon, Ayelet Gneezy.

**Writing – original draft:** Ariel Fridman, Rachel Gershon, Ayelet Gneezy.

**Writing – review & editing:** Ariel Fridman, Rachel Gershon, Ayelet Gneezy.

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
