## [Decision Letter · Decision Letter 0]

8 Jan 2021

PONE-D-20-35660

COVID-19 and Vaccine Hesitancy: A Longitudinal Study

PLOS ONE

Dear Dr. Fridman,

Thank you for submitting your manuscript to PLOS ONE. After careful consideration, we feel that it has merit but does not fully meet PLOS ONE’s publication criteria as it currently stands. Therefore, we invite you to submit a revised version of the manuscript that addresses the points raised during the review process.

Please find below the reviewer's comments, as well as those of mine.

We look forward to receiving your revised manuscript.

Kind regards,

Valerio Capraro

Academic Editor

PLOS ONE

Additional Editor Comments:

I have now collected one review from one expert in the field. I was unable to find a second reviewer. However, the one review I could collect is very detailed; moreover, I am myself familiar with the topic of this manuscript. Therefore, I feel confident in making a decision with only one review. The review is positive but suggests a major revision. I agree with the reviewer. Therefore, I would like to invite you to revise your work following the reviewer's comments. I only have one more comment, beyond those of the reviewer. You also look at the correlation between risk perception and vaccine hesitancy. I have recently published a paper where we look at the correlation between risk perception and intentions to wear a face mask. The results are in line with your study. I think it could be interesting to relate these works.

Capraro, V., & Barcelo, H. (2020). The effect of messaging and gender on intentions to wear a face covering to slow down COVID-19 transmission. Journal of Behavioral Economics for Policy.

Journal Requirements:

Reviewers' comments:

Reviewer's Responses to Questions

**Comments to the Author**

1. Is the manuscript technically sound, and do the data support the conclusions?

Reviewer #1: Yes

2. Has the statistical analysis been performed appropriately and rigorously? 

Reviewer #1: I Don't Know

3. Have the authors made all data underlying the findings in their manuscript fully available?

Reviewer #1: Yes

4. Is the manuscript presented in an intelligible fashion and written in standard English?

Reviewer #1: Yes

5. Review Comments to the Author

Reviewer #1: The manuscript describes a very interesting longitudinal study conducted on a sample of US citizens, regarding their attitudes towards vaccines in general and intention to get COVID-19 vaccine.

However, while the research itself, and its results, are very interesting and with potentially useful implications, I feel that the quality of the report needs to be improved, as I will outline in detail below.

Firstly and foremost, I find a bit awkward the the choice of the authors of introducing parts that should pertain to the discussion of the results in the introduction (i.e. lines 62-65 and 92-100).

The same goes for the results section: in this section, the authors included not only the results, but also a (a bit confused, in my opinion) explanation of data analyses. I recommend the authors to re-organize this section, adding a paragraph in "methods" to explain their analyses plan before describing results: an example (but not exhaustive) are lines from 148 to 153, as these are methods and not results. Another example are lines 169-172. Re-organizing these sections will greatly increase readability and clarity.

Regarding data analyses, I have some concerns: I was expecting an approach based on the analysis of variance (ANOVA), to address means differences within waves and between different groups. So, I'm not really sure that the approach adopted by the authors is the most suitable. However, I expect that the aforementioned re-organization of the methods and results sections will help (me and the future readers) to understand the authors' choices, and the authors to justify the methods they adopted.

Moreover, given sample size and the number of tested hypotheses, I would like the authors to address the fact that some of their "significances" were rather marginal (e.g. see p-value=.046 considered significant at line 188). I feel like the authors should address this by either adopting a more conservative value of p (instead of the usual p=.05), or by adding some note of caution in the discussion for those results that are only marginally significant. On the same page, I would like the authors to add effect sizes were applicable, e.g. Cohen's d (or similar) when reporting t-tests.

Finally, one last concern regarding the sample: were any strategies used to check data quality? Unfortunately, using data from panel results sometimes in some participants being "professional respondents": were any countermeasures taken (e.g. screening of multivariate outliers, uncommont response patterns or survey completion times)? if not, this should be address and discussed by the authors in the manuscript.

The bottom line is: the study is of great importance on a paramount topic. The results themselves are interesting, with potentially useful implications. It also has been conducted rigorously, although I would like some methodological choices to be explained better. However, the quality of the manuscript needs to be improved, in particular for what concerns the organization of the sections.

6. PLOS authors have the option to publish the peer review history of their article (what does this mean?). If published, this will include your full peer review and any attached files.

Reviewer #1: **Yes: **Lorenzo Palamenghi

---

## [Author Response · Author response to Decision Letter 0]

3 Feb 2021

February 3, 2021

COVID-19 and Vaccine Hesitancy: A Longitudinal Study

Dear Dr. Capraro,

Thank you for the opportunity to revise our manuscript, titled “COVID-19 and Vaccine Hesitancy: A Longitudinal Study” for resubmission to PLOS ONE. We greatly appreciated the constructive feedback we received. Below we have outlined our response to each point that you and the reviewer raised in the review.

AE: I have recently published a paper where we look at the correlation between risk perception and intentions to wear a face mask. The results are in line with your study. I think it could be interesting to relate these works.

Thank you for bringing this paper to our attention. We incorporated it into the general discussion, on lines 329-331.

R: I find a bit awkward the choice of the authors of introducing parts that should pertain to the discussion of the results in the introduction (i.e. lines 62-65 and 92-100). 

Thank you for your feedback on the layout of the introduction. We streamlined the introduction, as per your suggestion, and removed lines 62-65. We revised lines 92-100 as well, which constitute the last few lines of the introduction and transition to the empirical section. We believe that these lines provide an important preview of the results which may be helpful to readers seeking a quick summary. However, if you still feel that this way of introducing the results is inappropriate, we will change it.

R: The same goes for the results section: in this section, the authors included not only the results, but also a (a bit confused, in my opinion) explanation of data analyses. I recommend the authors to re-organize this section, adding a paragraph in "methods" to explain their analyses plan before describing results: an example (but not exhaustive) are lines from 148 to 153, as these are methods and not results. Another example are lines 169-172. Re-organizing these sections will greatly increase readability and clarity.

Based on your feedback, we have made changes throughout the methods, results, and discussions sections to increase clarity. For example, as the reviewer suggested we moved lines 148-153 and 169-172 to the methods section (now 140-145 and 119-125, respectively). We also included a paragraph describing our analysis plan. We believe that these edits have improved the manuscript. Thank you for the suggestion.

R: I was expecting an approach based on the analysis of variance (ANOVA), to address means differences within waves and between different groups. So, I'm not really sure that the approach adopted by the authors is the most suitable. However, I expect that the aforementioned re-organization of the methods and results sections will help (me and the future readers) to understand the authors' choices, and the authors to justify the methods they adopted.

We re-organized the methods and results section, and hope they are now clearer. In the newly added analysis plan paragraph (lines 140-148), we provided additional justification for the fixed-effects regression model we used to analyze our data. Furthermore, based on your suggestion, we added an ANOVA table with all results to the manuscript (lines 174-176, table S4).

R: Moreover, given sample size and the number of tested hypotheses, I would like the authors to address the fact that some of their "significances" were rather marginal (e.g. see p-value=.046 considered significant at line 188). I feel like the authors should address this by either adopting a more conservative value of p (instead of the usual p=.05), or by adding some note of caution in the discussion for those results that are only marginally significant. On the same page, I would like the authors to add effect sizes were applicable, e.g. Cohen's d (or similar) when reporting t-tests.

The point about P-values is well taken. We included a sentence (line 145) explicitly pointing out that the P-values reported are not adjusted for multiple testing. Part of the reason we decided not to adjust the P-values is due to the fact that there is not a broad consensus on the correct way to do this. However, all data is available to readers who would like to adjust the P-values in their preferred way. Furthermore, the particular P-value the reviewer mentioned, regarding the time trend for Republicans, is not pertinent to our claim of divergence between Democrats and Republicans over time (which are P < .001 on all the vaccination attitudes and intentions measures). We also added Cohen’s d effect sizes for all t-tests, as suggested, as well as 95% confidence intervals for the effect sizes.

R: Finally, one last concern regarding the sample: were any strategies used to check data quality? Unfortunately, using data from panel results sometimes in some participants being "professional respondents": were any countermeasures taken (e.g. screening of multivariate outliers, uncommon response patterns or survey completion times)? If not, this should be address and discussed by the authors in the manuscript.

While data from Amazon’s Mechanical Turk has advantages and disadvantages, with extant research examining the reliability and quality of this sample (e.g., Goodman, Cryder, and Cheema 2012), we have confidence in our choice to use this population for our study. We can further use our own data to demonstrate the quality of our sample. One indication that participants are paying attention is that the demographic makeup of our sample is stable over time, indicating that they did not respond to these questions at random. Due to your concerns, we ran an additional robustness check in which we removed participants with completion times below the 10th percentile, corresponding to less than 3 minutes. We found a similar pattern of results, though some coefficients were no longer significant at the .05 level, which is unsurprising given the smaller sample size. These include: overall decline in flu shot intentions (p = .06), Republican’ decline in COVID-19 vaccination attitudes (p = .10), Democrats’ decline in general vaccination attitudes (p = .08), Democrats’ increase in perceived threat of COVID-19 (p = .06). Importantly, the difference in trends between Democrats and Republicans remained significant in all cases. This robustness check and a robustness check that includes only participants who completed all 6 waves of the study can be easily run using our code. We would also argue that if the participant quality were low or if participants were not paying attention, this would add noise to the data, and work against finding significant results, rather than introduce a systematic bias. 

Reference: Goodman, Joseph K., Cynthia E. Cryder, and Amar Cheema. "Data collection in a flat world: The strengths and weaknesses of Mechanical Turk samples." Journal of Behavioral Decision Making 26, no. 3 (2013): 213-224.

Once again, we very much appreciate the in-depth feedback we received and think that the paper is much improved as a result of these changes. We hope that you will agree and we look forward to receiving your reply.

Sincerely,

Ariel Fridman

PhD Candidate in Marketing, Rady School of Management, UC San Diego

Rachel Gershon

Assistant Professor of Marketing, Rady School of Management, UC San Diego

Ayelet Gneezy

Professor of Behavioral Sciences and Marketing, Rady School of Management, UC San Diego

---

## [Decision Letter · Decision Letter 1]

31 Mar 2021

COVID-19 and vaccine hesitancy: A longitudinal study

PONE-D-20-35660R1

Dear Dr. Fridman,

We’re pleased to inform you that your manuscript has been judged scientifically suitable for publication and will be formally accepted for publication once it meets all outstanding technical requirements.

Kind regards,

Valerio Capraro

Academic Editor

PLOS ONE

Additional Editor Comments (optional):

Reviewers' comments:

Reviewer's Responses to Questions

**Comments to the Author**

1. If the authors have adequately addressed your comments raised in a previous round of review and you feel that this manuscript is now acceptable for publication, you may indicate that here to bypass the “Comments to the Author” section, enter your conflict of interest statement in the “Confidential to Editor” section, and submit your "Accept" recommendation.

Reviewer #1: All comments have been addressed

2. Is the manuscript technically sound, and do the data support the conclusions?

Reviewer #1: Yes

3. Has the statistical analysis been performed appropriately and rigorously? 

Reviewer #1: Yes

4. Have the authors made all data underlying the findings in their manuscript fully available?

Reviewer #1: Yes

5. Is the manuscript presented in an intelligible fashion and written in standard English?

Reviewer #1: Yes

6. Review Comments to the Author

Reviewer #1: In the first round I raised two different sets of comments: some on the methodology and others on the clarity of the report.

As for the methodological concerns I raised, I feel like the authors' resposes are adequate, and I have no further concerns to this regards.

As for the clarity, I think that the reorganization of the various sections has much improved the quality of the manuscript and the overall clarity. Still, the authors have decided to leave a small "anticipation" of the results (which, to me, really looks like a discussion of the results) in the introduction.

At this point, I think it comes to a matter of personal preferences: personally, I believe that the abstract should give the readers a quick summary, and that the introduction should introduce (and not anticipate or summarize) the presented study. However, those few lines do not impact the overall clarity, so I won't ask for further revisions and I'll let the authors (or the editor, eventually) decide whether these lines should be changed or not, as this in my opinion goes beyond the scope of the peer review: the study is solid, interesting, quite well reported, and should be accepted for publication.

7. PLOS authors have the option to publish the peer review history of their article (what does this mean?). If published, this will include your full peer review and any attached files.

Reviewer #1: **Yes: **Lorenzo Palamenghi

---

## [Editor Report · Acceptance letter]

7 Apr 2021

PONE-D-20-35660R1 

COVID-19 and vaccine hesitancy: A longitudinal study 

Dear Dr. Fridman:

I'm pleased to inform you that your manuscript has been deemed suitable for publication in PLOS ONE. Congratulations! Your manuscript is now with our production department. 

Kind regards, 

on behalf of

Dr. Valerio Capraro 

Academic Editor

PLOS ONE